# Some Specifics of Defect-Free Poly-(o-aminophenylene)naphthoylenimide Fibers Preparation by Wet Spinning

**DOI:** 10.3390/ma15030808

**Published:** 2022-01-21

**Authors:** Ivan Y. Skvortsov, Valery G. Kulichikhin, Igor I. Ponomarev, Lydia A. Varfolomeeva, Mikhail S. Kuzin, Dmitry Y. Razorenov, Kirill M. Skupov

**Affiliations:** 1A.V. Topchiev Institute of Petrochemical Synthesis of Russian Academy of Sciences, Leninsky Av. 29, 119991 Moscow, Russia; klch@ips.ac.ru (V.G.K.); varfolomeeva.lidia@mail.ru (L.A.V.); gevahka15@gmail.com (M.S.K.); 2A.N. Nesmeyanov Institute of Organoelement Compounds of Russian Academy of Sciences, Vavilova St. 28, 119991 Moscow, Russia; gagapon@ineos.ac.ru (I.I.P.); razar@ineos.ac.ru (D.Y.R.); kskupov@gmail.com (K.M.S.)

**Keywords:** poly-(o-aminophenylene)naphthoylenimide, heterocyclic polyarylenes, polymer solution rheology, coagulation, wet spinning, fibers

## Abstract

A series of model experiments were carried out on drops of poly-(o-aminophenylene)naphthoylenimide (PANI-O) solutions in N-methyl-2-pyrrolidone (NMP) surrounded by a coagulant of different compositions as starting points of defect-free fibers spinning by the wet method. An influence of compositions of dopes and multicomponent coagulants on the diffusion kinetics and drop morphology during coagulation has been investigated. It is shown that the defining parameters of the coagulation process are viscoelastic properties of the polymer solution and the diffusion activity of the coagulant, meaning not only the rate of coagulation but also the presence/absence of macro defects in the resulting fiber. The optimal morphology of as-spun fibers is obtained by coagulation of solution in a three-component mixture containing solvent and two precipitants of different activity (water and ethanol). The chosen coagulating mixture was used for the fiber spinning of PANI-O with different molecular weights dopes, and fibers with sufficiently high strength (~250 MPa), moduli (~2.1 MPa), and elongation at break (50%) were obtained.

## 1. Introduction

Fire- and heat-resistant semi-ladder polynaphthoylbenzimidazole (PNBI) has one of the highest thermal, heat-, fire-resistance properties and limiting oxygen index among all organic polymers [1,2,3]. Application of this polymer is limited only by solubility only in strong acids; also, the low solution concentration for spinning or film casting determines the high cost of the product and makes that process ecologically unfriendly [4]. PNBI has a general formula shown on Figure 1 [5].

The chemical structure of PNBI includes six isomeric elements due to the possibility of the existence of “cis-“ and “trans-“ isomers in the condensed system, as well as the growth of the polymer chain through the meta- or a para- positions of the amino groups of the starting tetraamine. PNBI fibers, traditionally prepared from solutions in inorganic acids [4,6,7], can be spun via the environmentally friendly and economically feasible method by heat treatment of fibers based on poly(o-aminophenylene)naphthoylenimide (PANI-O) precursor [8] (Figure 2).

In contrast to PNBI, synthesized in polyphosphoric acid at temperatures of 160–200 °C, PANI-O could be synthesized in organic solvents like dimethyl sulfoxide (DMSO) or N-methyl-2-pyrrolidone (NMP) at room temperature [5].

The solubility and rheological parameters of PANI-O solutions in organic solvents continuously change in time due to the occurrence of parallel reactions of intramolecular cyclization and polymerization [9]. The optimal level of rheological properties and several preliminary experiments have shown that the wet-spinning of these solutions is the most favorable for obtaining qualitative fibers [8,9]. For this method, the composition of the coagulation bath is the most important parameter from the viewpoint of obtaining defect-free fibers [10,11]. The morphology of the latter, namely the shape of the cross-section, radial gradient of ordering due to the core-shell structure, and presence of macro- and micropores are mainly predetermined by the coagulation conditions. The composition and the temperature of the coagulation bath, as well as the rheological properties of the spinning solutions themselves, are the key parameters responsible for the rate of interdiffusion of the solvent and coagulant in the coagulation process [12].

The simple one-component coagulant is likely the most preferable, but from a technological point of view, this approach is acceptable in some rare cases, for example, in the dry-wet spinning of cellulose solutions in a crystalline solvent—N-methyl morpholine-N-oxide (NMMO) [13,14]. When jets of high-viscosity hot solution from the air gap enter into cold water (one-component coagulant), two processes occur simultaneously: solution crystallization and diffusion mass transfer washing out of the solidified solvent, leading to concentrating the polymer reached phase. These processes prevent the formation of defects and result in the monolithic morphology of fibers. However, even for such a system, the appearance of defects in the coagulation process with an increase in the temperature of the coagulant was found. In principle, it may be stipulated by the viscosity ratio of the coagulant and the dope, which also is studied in the present paper.

However, using a one-component coagulant causes serious problems in technological processes, since a solvent is spent almost completely. It is much better to use mixtures of coagulant and solvent, as in the production of PAN fibers from solutions in aprotic solvents [15,16,17]. Such mixtures pass much easily through the regeneration procedures and a large part of a solvent comes back to the technological process. Moreover, in the technology of the above-mentioned cellulose fibers production usually, mixtures of the solvent and coagulants have been used.

For PANI-O, due to the change of its structure in time and the relatively low viscosity of spinning solutions, an application of a one-component coagulant leads to the formation of a large number of defects [8]. One of the prospective ways to solve the problem of inhomogeneous coagulation is the introduction of a non-solvent into the dope [9,18,19]. In the above-mentioned publications, this approach helped to reduce significantly the number of radial finger-like defects in the fiber during the coagulation process. In addition, these studies show that the presence of a certain amount of a non-solvent in the solution promotes structuring, accompanied by an increase in viscoelastic properties to a level optimal for stable spinning.

Thus, to obtain high-quality heat and fire-resistant PNBI fibers with high limiting oxygen index from PANI-O solutions in NMP, it is necessary to keep in mind a combination of factors: change of the polymer solution properties over time, which affects the complexity of the properties of solutions (among them the rheological are the main ones); methods of spinning and composition of coagulant allowing organization of the defect-free spinning. This study is of particular interest for the development of membrane gas separation technology based on PNBI [20]. This paper presents the results of a systematic study of the influence of these factors on the coagulation processes of PANI-O solutions and, as a result, on the quality of the fibers being spun. This approach is fundamentally important in developing a strategy for spinning fibers from a new polymer and insufficiently studied solutions.

## 2. Materials and Methods

### 2.1. Materials

#### 2.1.1. Monomers and Solvents

3,3,4,4-tetraaminodiphenyl ether (Rubezhnoe plant, Rubezhnoe, Ukraine) with a melting point of 155–156 °C and 1,4,5,8-naphthalenetetracarboxylic acid dianhydride (VNIPIM, Tula, Russia) were dried at 100 °C under vacuum before use. Benzimidazole, benzoic acid, and NMP (Thermo Fisher Scientific, Acros Organics, Waltham, MA, USA) were used as received.

#### 2.1.2. Synthesis of PANI-O

Total of 2.3027 g (0.01 mol) of o-tetraaminodiphenyl ether, 2.6818 g (0.01 mol) of 1,4,5,8-naphthalenetetracarboxylic acid dianhydride, 0.24 g (0.00196 mol) of benzoic acid, 0.24 g (0.002 mol) of benzimidazole, and 20 mL of anhydrous NMP were placed in the flask and stirred in Ar flow for 24 h at room temperature forming a viscous solution of polyaminonaphthoyleneimide (PANI). The reaction mixture was then stirred at 50–60 °C until the agitator Heidolph (Schwabach, Germany) was almost stopped due to very high viscosity. This method was used to obtain polymer samples with an intrinsic viscosity of 0.6–0.8 dL/g and solution concentrations from 10 to 15%. Further growth of the molecular weight of the samples was monitored by the isothermal holding of the solutions in the operating unit of the rotary rheometer HAAKE MARS 60 (Thermo Fisher Scientific, Karlsruhe, Germany) at a given temperature, evaluating the change in viscoelastic properties, which made it possible to reach their level required for the successful fibers spinning.

#### 2.1.3. PANI-O Solutions

Solutions of PANI-O with various molecular weights were prepared by controlled heating of reaction system at 70 °C under permanent stirring with a J-shaped stirrer rotating at a speed of 10 rpm. As a result, a series of solutions with different heating times were obtained, and the intrinsic viscosity was determined by their dilution and measuring in Ubbelohde viscometer (Moscow, USSR) at 25 °C according to ASTM D2857 [21]. The molecular characteristics of polymers in the test solutions are presented in Table 1.

### 2.2. Methods

#### 2.2.1. Selection of a Coagulant

Morphology evolution of solution droplet (a model of the jet/fiber cross-section) surrounded by a coagulant was studied by a previously developed method [15,23]. The experimental setup was mounted on the bright-field optical microscope, equipped with a camera. A drop of solution was placed between slide and cover glass, and the prepared cell was transferred on the microscopy stage plate (Biomed Co, Moscow, Russia,) (Figure 3, right). Then, a drop was surrounded with a coagulant, and the resulting morphology was observed on a computer monitor (Figure 3, left).

A detailed scheme of a measuring cell is shown in Figure 4.

The experiment was carried out as follows. A drop of a solution with a volume of ~0.1–0.2 mm^3^ was put onto a slide from a syringe. The sample was immediately closed with a cover glass, and a coagulant was added to the gap between glasses, which, due to capillary forces, quickly flowed inward, surrounding a drop of a polymer solution. For the correct experiment, the volume of the drop must be at least 100 times less than the volume of the added coagulant. This condition is necessary to exclude a dilution of the coagulant with the solvent released from the droplet as a result of the diffusion stream. The thickness of the droplets and their average diameter were ~0.1 mm and ~1.5 mm, respectively. Since these solutions upon contact with ambient moisture form gel easily, the experiments were carried out in a dry box (handmade, Moscow, USSR) with a relative humidity of less than 1%. Then the cell was taken out from the box and the morphology of the coagulated drop was examined.

In the case when it is necessary to investigate the kinetics of the coagulation process [9], the experiment was proceeded on the microscope stage, using an isolated transparent measuring cell, to exclude change of composition because of evaporation or, vice versa, wetting by the air moisture. The observation of polymer solution-coagulant interaction was fulfilled on the Biomed 6PO microscope (Biomed Co., Russia, Moscow). Tests for all coagulants were performed at 25 °C. 

The following liquids as potential coagulants for the wet spinning process were tested: water, ethanol, propanol-1, butanol-1, heptanol-1, octanol-1, propylene glycol, glycerol, and their mixtures with NMP. All these reagents with a purity of ≥99% were supplied from Ecos-1 (Moscow, Russia).

#### 2.2.2. Rheology

The rheological properties of solutions were measured in steady-state and oscillation regimes of the shear strain were determined using rotational rheometer HAAKE MARS 60 rheometer (Thermo Fisher Scientific, Karlsruhe, Germany) at the 25 °C. The behavior of the solutions was tested using a cone and plate unit with a cone diameter of 60 mm and an angle between cone and plate of 1° being used for the tests at different shear rates and temperatures.

The flow curves were recorded at the shear rate range of 10^−2^–10^4^ s^−1^. The frequency dependences of the storage and loss moduli were measured in the angular frequency range of 0.6–628 rad/s in the linear viscoelastic range at a strain of 1%.

#### 2.2.3. Wet Spinning

Fibers from PANI-O solutions with different viscoelastic characteristics were obtained on a laboratory wet-spinning stand (handmade, Russia, Moscow), schematically shown in Figure 5. When developing the spinning regimes, the data of modeling the coagulation on a drop of solutions and their rheological properties were kept in mind, since they are the key factors that directly affect the outflow of spinning solutions from the spinneret channels, as well as the morphology and mechanical properties of the spun fibers.

The fibers were spun from solution flowed out of a syringe with a constant feeding rate through a spinneret with 100 holes of 80 μm in diameter. Various types of coagulants selected from the results of the droplet coagulation experiments were used. The draw values on different stages were chosen based on a provision of the spinning stability.

#### 2.2.4. Fiber Characterization

The diameter of each fiber was determined as the average of ten measurements at different locations along with the fiber. To perform these measurements, the Biomed 6PO optical microscope coupled with the ToupTek Photonics XFCAM1080PHD (Hangzhou, China) camera with 60× magnification lens was used. The accuracy of measurements was ±0.3 µm. For every thread at least 10 filaments were examined. Inhomogeneity was characterized by the difference between the maximum and minimum values of the fiber diameter.

Mechanical properties of PANI-O fibers were measured using the Instron 1122 (Norwood, MA, USA) tensile machine at a basic filament length of 10 mm and an extension rate of 10 mm/min. All measurements were performed at 23 ± 2 °C. The reported results were averaged for at least ten tests.

## 3. Results and Discussion

### 3.1. Rheology

The solution’s rheology with varying PANI-O concentrations and molecular weights is described in detail earlier [9]. In part, the former data were compared with those obtained in this work. The flow curves and frequency dependences of dynamic moduli obtained in the range of linear viscoelasticity for all samples numerated in Table 1 are presented in Figure 6.

Solution P1 is a standard viscoelastic fluid with Newtonian behavior in the entire investigated range of shear rates and well obeying the power-law frequency dependences of elastic and loss moduli in the terminal zone with exponents ~2 and 1, respectively.

An increase in the intrinsic viscosity from 0.6 to 0.9 dL/g (sample P2) results in a significant effect on the rheological properties of the solution—the region of the Newtonian viscosity disappears, and the slopes of the logarithmic frequency dependences of the moduli decrease to 1 for G′ and 0.7 for G″. An increase in the concentration of the polymer (solution P3) leads to an increase in the absolute values of the moduli and viscosity, while their dependences on frequency or shear rate remain unchanged.

A further increase in the molecular weight (samples P4 and P5) leads to a significant structuring of the systems. The exponent values of the frequency dependences of the storage and loss moduli become close to ~0.6.

The selected series of solutions made it possible to carry out a systematic study of the effect of various coagulants on the model solutions coagulation in a wide range of their viscoelastic characteristics.

### 3.2. Coagulation

Polymer separation from the solution in the wet spinning process begins with the separation of the solution into concentrated and diluted phases. It is caused by a change in the polymer–solvent interaction parameter upon contact with a nonsolvent (precipitant or coagulant) [14]. It is customary to divide coagulants into stiff and soft according to their activity of interaction with the solution [24]. Using the stiff, i.e., coagulants that quickly diffuse into the jet of the solution, is not the best choice, since a high diffusion rate can lead to the formation of a “shell-core” morphology, a large number of defects, and significant unevenness of the obtained fibers. The softening of the coagulation process is usually achieved with an increase in the affinity of the coagulant for the solvent [25] and this is realized either by diluting the stiff coagulant with a solvent or by reducing the interdiffusion rate of the polymer solution and the coagulant by increasing the viscosity of the diffusing components.

In this work, the main attention was paid to the selection of a suitable coagulation composition for a specific polymer solution to obtain further fibers by the wet spinning method. As the first step, let us consider the results of the various factors influence that determine the coagulated PANI-O solution droplet morphology.

#### 3.2.1. Nonsolvent Nature Influence

An example of typical coagulation of a PANI-O solution droplet with a “stiff” coagulant (NMP/water mixture with a 70/30 components ratio) is shown in Figure 7. The processes of diffusion interaction and coagulation proceed according to two mechanisms: (1) large-scale (convective) penetration of the coagulant into the droplet with the formation and growth of vacuoles (Figure 7a and Appendix A), and (2) small-scale (diffusion) decomposition of the solution into concentrated and dilute phases with the movement of the deposition front from the droplet surface and from the vacuole boundaries to the center of the solution droplet (Figure 7b). 

Initially, a single-phase solution in the coagulation process converts into dispersion, which retains its continuity, and a disperse phase with a droplet/particle size (judging by their nonspherical shape) of the order of 1–2 μm.

When a drop of solution contacts with the coagulant, a gel-like shell is formed around the perimeter of the drop, sooner its edge, and in the case of a “stiff” coagulant and a low-viscous solution, it breaks due to a large difference in the concentration of the coagulant on the surface and inside the drop. The coagulant penetrates inside through cracks in the shell, forming vacuoles in the volume. Further, the process of phase separation proceeds in the following way. Inside the vacuoles the concentration of the nonsolvent is much higher than in the surrounding solution, which is why the diffusion should proceed through the interfaces of vacuoles with predominant direction from vacuoles into drop’s volume. Vacuoles present in the as-spun fibers prepared using “stiff” coagulant, transform to the hollow inclusions in the final fibers.

So, the main feature of using the “stiff” coagulant consists in the decisive role of the vacuoles walls as membranes regulating the interdiffusion process. Because of the penetration of the coagulant from vacuole spaces into the solution, the critical value of the Flory-Huggins parameter is achieved, and according to rules of the amorphous phase equilibrium, described by binodal line, the neat solution decomposes on dilute and concentrated by polymer phases. Fractions of each phase depend on the location of the figurative point and can be estimated by the so-called “the lever rule”. According to Papkov [26], if the dilute phase grows faster than the concentrated one, the complete phase separation cannot be reached in real-time, and the formed gel can be characterized as a system with non-complete phase separation. In the opposite case, when the concentrated phase grows faster compared with the dilute one, a deposition or precipitation of the concentrated phase proceeds, leading to almost complete phase separation.

For a fiber spinning technology from polymer solutions, the first route, i.e., gel-like or as-spun fiber formation, has to be realized. In the case of modeling the spinning process on an unmoved drop, an indication of gel formation could be mechanical testing of coagulated drop and its more-less homogeneous morphology. But in the case of the “stiff” coagulant, the decisive role in the mass-transfer processes belongs to vacuoles interfaces because the phase separation takes place outside the vacuoles under the action of diffusion stream of coagulant from vacuoles. The kinetics of phase separation in different parts of the model droplet can be different, which gives rise to non-uniformity of the composition along the droplet diameter, considered as an analog of the cross-section of the spinning fiber.

Despite the absence of polymer inside vacuoles, the interdiffusion of coagulant from vacuoles and solvent from solution into vacuoles takes place. As the concentration of the nonsolvent in the vacuoles decreases, due to dilution with a solvent from the droplet volume, the growth of vacuoles stops and the discussed above phase separation mechanism begins to prevail. As a result, a large number of non-spherical particles/droplets in the volume of the model droplet forms. At the transition of a homogeneous solution into a heterogeneous system, the front of the coagulation is indicated by an arrow in Figure 7b, spreading along the drop radius. Upon completion of interdiffusion, a gel-like drop with a rather complex morphology is formed. It may contain a network of micro defects and vacuoles, distributed in a highly heterogeneous system. This process is the most clearly manifested in low-viscosity PANI-O solutions.

The combination of physicochemical (affinity between solvent and coagulant, diffusion penetration, mechanism of the phase separation) and rheological (viscosity, elasticity) factors has a significant effect on the interaction between solution jet and coagulant causing defect-less morphology of the fibers obtained. The best way consists in the realization of the homogeneous phase separation without large defects formation as shown in Figure 7b. Below, the role of individual factors and their combinations in modeling the wet-spinning process are analyzed. The main aim is to find the appropriate coagulant for almost unknown solutions of the relatively new polymer.

##### One-Component Coagulants

Water Coagulation

The simplest and cheapest coagulant for PANI-O solutions in NMP is water. This coagulant is the “stiffest” due to its high affinity to the solvent. It can be seen clearly from Figure 8 and some other coagulated solutions on Appendix A, where vacuoles of different morphology are evident. Because of the formation of a large number of defects, the use of water for fiber spinning is not of practical interest.

Alcohols Coagulation

Alcohols have a lower affinity to NMP, so they have to act as softer coagulants compared with water, but morphology depends significantly on the nature of their aliphatic part (Figure 9).

Analysis of the micrographs allows us to conclude that an increase in the length of the aliphatic fragment of alcohol reduces the number of large defects-vacuoles and causes more intense “diffusive coagulation” of the solution, that leads to the formation of an opaquer solidified drop. The main reason for the opacity is the presence of a large number of small particles. This feature of the coagulated drop can be caused by the anisotropic shape of inclusions.

##### Two-Component Coagulants

A well-known and simple way to slow down the diffusion and coagulation processes to make them more homogeneous is using the mixture of a coagulant and a solvent. This approach is widely applied in the field of spinning, for example, PAN fibers by the wet method [15,16]. The P2 solution as the representative sample was tested for coagulation with mixtures of NMP and water (Figure 10a1–a4) and ethanol (Figure 10b1–b4) in different ratios.

Introducing 30% of water to NMP (Figure 10a1) results in coagulation with a large number of defects. A decrease in the fraction of water to 20% (Figure 10a2) causes rapid coagulation with noticeably fewer large-scale defects. In such a system, the growth of vacuoles stops at the intermediate stage. At 17% of water, some fraction of the solution (apparently, low molecular weight polymer) begins to be dissolved in the coagulant. As can be seen in Figure 10 (Figure 10a3), it leaks out from the drop volume as a part of the coagulated particles, concentrates at its periphery, and even passes into the complex coagulant phase.

The selective disintegration/dissolution process is shown in detail in Figure 11. Under the action of this coagulant, the partially disintegrated edge of the droplet is observed with further formation of a large number of particles. In this case, no defects are formed in the droplet, however, such a soft coagulant violates the solidity of the coagulated sample periphery since this part is partially disintegrated and partially dissolved. The coagulant containing 15% of water (Figure 10a4) does not cause the phase separation of the solution at all, since the entire drop dissolves, which proves the compatibility of solutions with a soft non-solvent, as was shown in [9].

Ethanol is a softer coagulant than water, but when its content in NMP is 50% (Figure 10b1), vacuoles formation and a pronounced phase separation inside and outside of them are observed in the drop. The indicator of the phase separation is the blackening of the heterophase areas. A yellowing of the periphery of the solution droplet indicates an insignificant wash-out of the soluble fraction. At the ethanol content of 40% (Figure 10b2) judging by the weak darkening of the central part of the droplet, the phase separation proceeds less intensively, although the number of large defects remains high. A composition with the alcohol concentration of 30% (Figure 10b3) no longer initiates coagulation of the solution drop (it remains liquid during the observation time), and a further decrease in the alcohol fraction to 20% (Figure 10b4) leads to the complete dissolution of the drop. In the homologous series of alcohols, a slight decrease in the degree of defectiveness of the coagulated droplet starting with butanol is observed as could be seen in Appendix A.

To summarize, we can conclude that the softening of the coagulation activity of water or alcohols by solvent leads to a decrease in the number of defects. However, its complete disappearance proceeds at such a concentration of the solvent, which does not cause coagulation. That is why the applicability of these two-component coagulants is excluded.

##### Three-Component Coagulants

Evaluation of the effect of water and ethanol on the coagulation of PANI-O solutions in NMP renders it possible to suggest the optimal composition for three-component NMP/water/alcohol coagulants. Such a composition initiates coagulation by water and reduces the number of vacuoles by including ethanol. Figure 12 shows a selected series of solution drop images after coagulation by ethanol/water/NMP with different component ratios.

Ratios between components of coagulants were chosen according to the following considerations. An increase in the NMP concentration above 70% at all analyzed contents of water and ethanol transforms the coagulant into a solvent. A decrease in the fraction of NMP below 65% leads to an appearance of defects. At an NMP concentration of 65–70%, an increase in the water fraction above 10% with a corresponding decrease in the ethanol content (Figure 12b) leads to the formation of large defects, the size and number of which increase along with the water content (Figure 12a). The same effect is observed with an increase in the portion of ethanol (Figure 12d). As a result, in the course of this expertise, a narrow concentration range of the three-component composition was found (Figure 12c), which allows obtaining defect-free coagulated drops of PANI-O solution. This area is located on a coagulation-dissolution border and is more clearly shown in the three-component diagram (Figure 13).

It should be noted that the diagram shows the region of the coagulant compositions which makes it possible to obtain defect-free coagulated samples from a solution of polymer with an intrinsic viscosity of 1.2 dL/g and a shear viscosity of ~20 Pa·s. These parameters (the viscosity of the solution and the composition of the coagulant) can be considered minimally sufficient for coagulation without the formation of the large defects—vacuoles. As will be shown in the next section, that the low viscosity of solution leads to the formation of defects in any range of compositions, and vice versa, an increase in the molecular weight of the polymer (and, accordingly, the viscosity of the solution) expands the range of the coagulant compositions that does not cause the defects formation.

#### 3.2.2. Coagulant Viscosity Influence

To assess the effect of the interdiffusion kinetics on the coagulation processes, some alcohols with different viscosities were selected: ethanol (~0.001 Pa·s), ethylene glycol (~0.02 Pa·s), and glycerol (~1.5 Pa·s). In general, the coagulant viscosity determines the diffusion penetration kinetics into solution drop, as well as its inertia, which is reflected in the duration of complete coagulation of the same size drop. An attempt to test as coagulants alcohols with one (ethanol), two (ethylene glycol), and three OH-groups (glycerol) with different viscosity did not lead to any change in many large-scale defects in the coagulated drop (Appendix A) despite their viscosity differences up to three orders. 

A more important factor should be the concentration of –OH groups in alcohol molecules that also could affect the affinity of coagulant to solvent. One could bear in mind the possibility of interaction of the –OH groups with the oxygen of the lactam ring leading to the formation of H-bonds and deactivation of NMP as a PANI-O solvent. However, from viewpoint of obtaining a defect-free morphology of the coagulated droplet, both of these factors do not play a decisive role in the kinetics of diffusion and coagulation.

#### 3.2.3. Solution Viscosity Influence

Another way to reduce the components interdiffusion rate is to increase the viscosity of the polymer solution, which can be realized in the isothermal mode only in two ways—by increasing either the polymer concentration or its molecular weight.

##### Polymer Concentration Influence

The polymer concentration in a drop of the solution has a significant effect on the coagulation kinetics. The corresponding data are presented in Figure 14.

At low concentrations, when the polymer solution viscosity is about 1 Pa·s (Figure 14a), the coagulation process proceeds rapidly with the formation of fragments of the coagulated solution separated by a liquid phase (diluted solution and a mixture of solvent and coagulant). An increase in the polymer concentration to 9% and viscosity, respectively, to 10 Pa·s, reduces the rate of interdiffusion, leading to the formation of a denser structure with fewer defects. Solution with a concentration of 12% and viscosity of about 80 Pa·s becomes the most suitable for homogeneous coagulation since the large-scale defects are not formed as a whole. Based on the data obtained, it can be concluded that solutions with a viscosity of ~80 Pa·s and a little higher are optimal for the wet-spinning process. The upper limit of solution concentration is determined by viscoelastic properties, in particular, its ability to viscous deformation when flowing from the die and elasticity, providing stable jets formation capable to be oriented at stretching.

##### Polymer Molecular Weight Effect

The influence of the dopes viscoelastic properties was tested on a series of solutions with a concentration of 10% containing polymers of various intrinsic viscosity. The growth of the molecular weight was reached by the controlled heating process. The morphology of the coagulated droplet is shown in Figure 15 using the typical binary and ternary coagulants.

Two-component water/NMP coagulant at optimal ratio of 20/80 makes it possible to obtain defect-free samples from a viscous high-molecular solution P5 only (Figure 15a). The most suitable is a three-component mixture consisting of water, ethanol (or n-propanol), and NMP, which promotes uniform coagulation of the solution droplet without large-scale defects, as is seen from the images in Figure 15b for solutions of medium (P4) and high (P5) molecular weights.

Based on the results obtained and conclusions made, it can be expected to obtain homogeneous and defect-free fibers from PANI-O solutions by a wet-spinning method. The polymer must have sufficiently high molecular weight, and the polymer solution has certain viscoelastic properties. The best coagulants for PANI-O solutions in NMP are ternary mixtures based on alcohols (ethanol/isopropanol/n-propanol), water, and NMP in a ratio of 20/10/70.

## 4. Fiber Spinning

Two solutions of PANI-O were selected to investigate the influence of the viscoelastic properties on the wet spinning process and to check the relationship between droplet coagulation experiments and processing by wet spinning. Solution P2 of relatively low molecular weight polymer with [η] = 0.9 dL/g manifesting Newtonian behavior and viscoelastic solution P5 of polymer with [η] = 1.8 dL/g have been coagulated by a mixture of ethanol, water, and NMP. The results of the drops coagulation of the selected solutions are shown in Figure 16.

As is seen, according to the morphology of the coagulated droplets both solutions are coagulated rather well under the action of chosen coagulants. For the polymer solution P2 (Figure 16a), few vacuoles are observed during coagulation by the composition ethanol/water/NMP = 25/10/65. At the same time, in a slightly softer coagulant, composed by changing the ratio between ethanol and NMP, the drop starts to dissolve (Figure 16b). But the same mixture works very smoothly at coagulation of the solution P5 (Figure 16c), leading to homogeneous morphology. 

Based on a model experiment, wet spinning of the above-mentioned compositions was performed to the three-component coagulant and fiber samples were obtained, images of which are shown in Figure 17.

A stable spinning process was realized for both solutions P2 and P5. In the case of a low-viscous solution (P2), the coagulant, which forms a minimum of defects (ethanol/water/NMP of the ratio 20/10/70) (Figure 16b), partially dissolves the fiber, making the spinning process impossible. For this reason, fibers were obtained using a “stiffer” coagulation composition (ethanol/water/NMP of the ratio: 25/10/65). Solution P5 is a structured viscoelastic system with close values of elastic and loss moduli in the entire frequency range (Figure 6b), and this behavior allows obtaining sufficiently high-quality fibers in the selected three-component coagulant. The mechanical properties of the resulting fibers are shown in Table 2.

The maximum mechanical characteristics were obtained for fibers formed from solution P5, and this result was achieved even for more thick fibers. Presumably, the implementation of higher draw ratios could lead to the production of fibers with higher mechanical characteristics.

## 5. Conclusions

Modeling the coagulation process with varying the coagulant activity (softness) and their viscosity, on the one hand, and the viscosity and viscoelasticity of the polymer solution, on the other, renders it possible to choose the most favorable coagulation parameters. As a result, the optimal coagulation composition and the complex of the polymer solution properties were selected for spinning the defect-free fibers. The key factors are the following: the certain viscoelasticity of the polymer solution, which stabilizes the jet formation, and the appropriate coagulant softness, which prevents the formation of defects during solution coagulation.

The correspondence of the model experiments and the real spinning situation was proved on a series of polymer solutions with different viscoelasticity and coagulation bathes based on NMP, water, and ethanol, mainly by varying the fractions of the last two components. The most suitable solution was spun to the chosen coagulant based on model experiments and the defect-free fibers with a strength of 250 MPa and elongation at a break of 50% were obtained. These mechanical properties are sufficient for their further thermal treatment for preparing fire-resistant polynaphthoylenebenzimidazole (PNBI) fibers.

## Figures and Tables

**Figure 1 materials-15-00808-f001:**
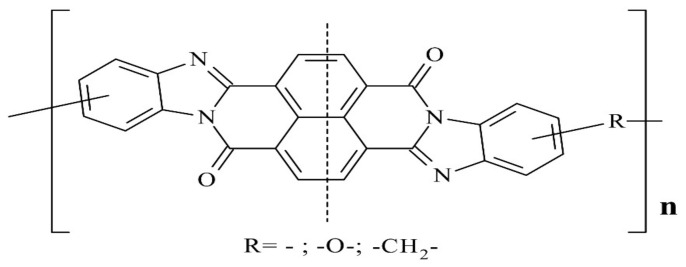
Structure formula of polynaphthoylbenzimidazole.

**Figure 2 materials-15-00808-f002:**
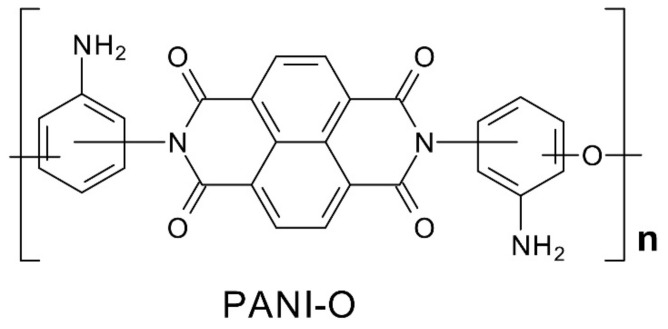
Structure formula of the poly(o-aminophenylene)naphthoylenimide (PANI-O).

**Figure 3 materials-15-00808-f003:**
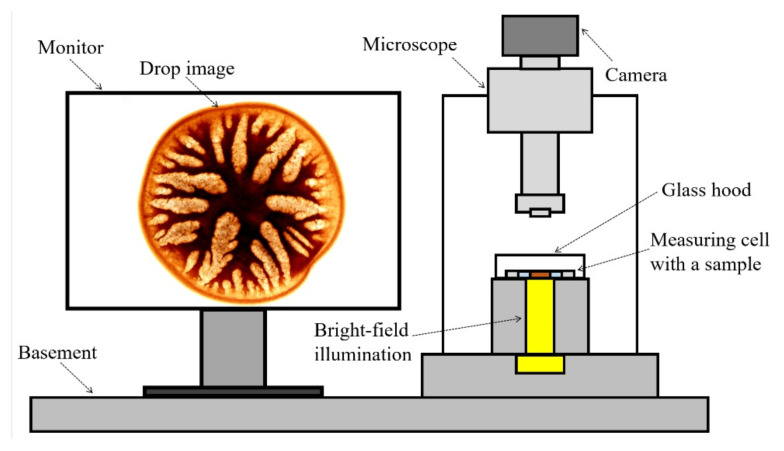
Scheme of the experimental setup.

**Figure 4 materials-15-00808-f004:**
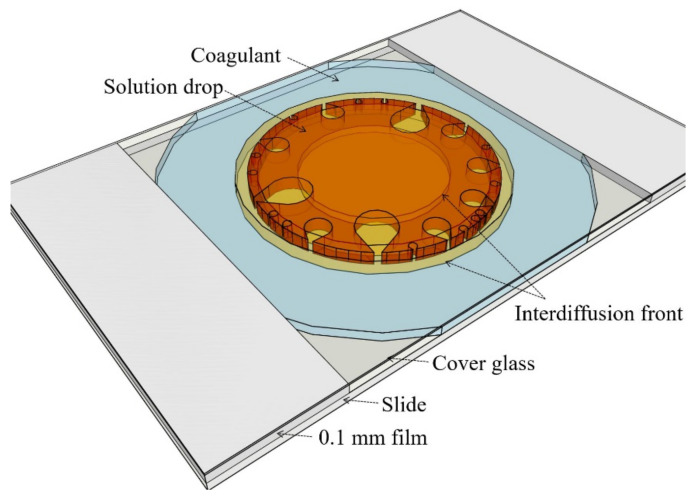
Scheme of the measuring cell with a combined sample (solution drop surrounded with coagulant).

**Figure 5 materials-15-00808-f005:**
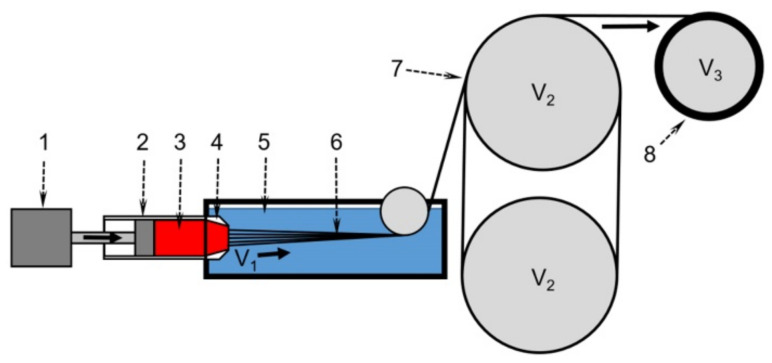
The scheme of the laboratory stands for wet spinning. 1—linear displacement device, 2—metallic syringe with a volume of 10 mL, 3—spinning solution, 4—multi-holes spinneret; 5—coagulation bath, 6—as-spun fibers, 7—pre-winding rollers, 8—winding roller at a speed V_3_. V_1_—the linear velocity of the solution flowing from the die, V_2_—the take-up velocity. V_2_ = V_3_.

**Figure 6 materials-15-00808-f006:**
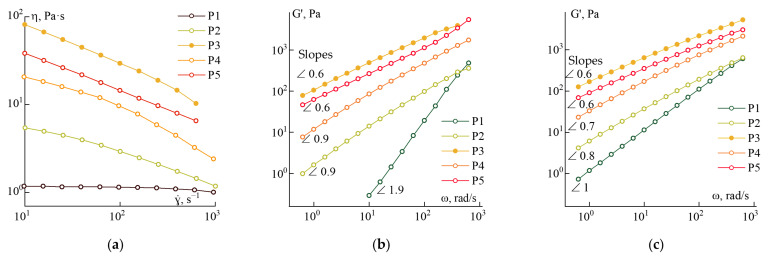
Flow curves (**a**), frequency dependences of elastic (**b**), and loss modulus (**c**) of the investigated solutions.

**Figure 7 materials-15-00808-f007:**
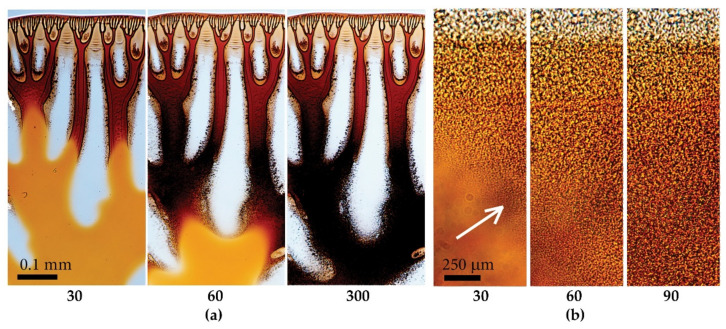
Coagulation kinetics of a PANI-O solution under the action of the N-Methyl-2-pyrrolidone (NMP)-water coagulant in a ratio of 70/30 using the example of P4. (**a**)—4× magnification; (**b**)—60× magnification. The arrow shows the phase separation boundary. The coagulation time from the beginning of the coagulant and solution drop contacting, in seconds, is shown at the bottom.

**Figure 8 materials-15-00808-f008:**
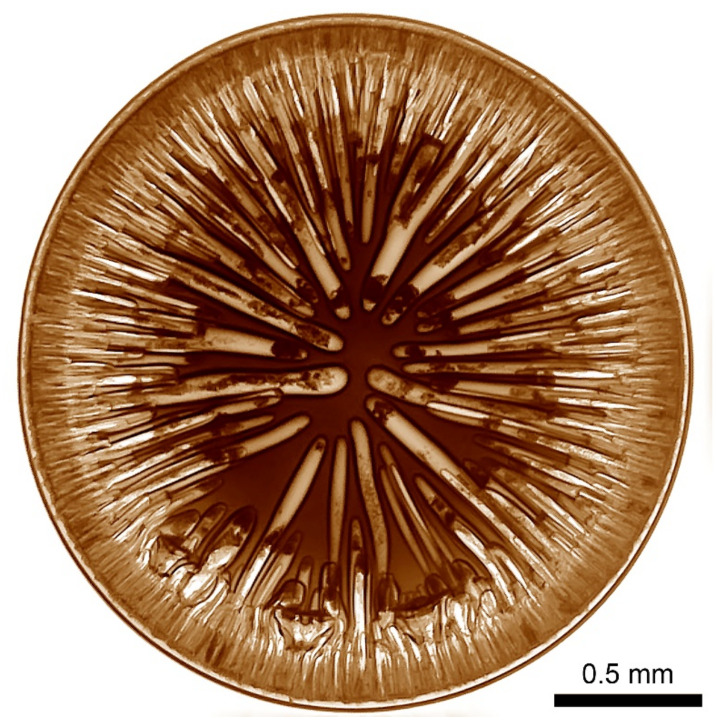
Coagulation of a PANI-O solution drop with water using the example of P3.

**Figure 9 materials-15-00808-f009:**
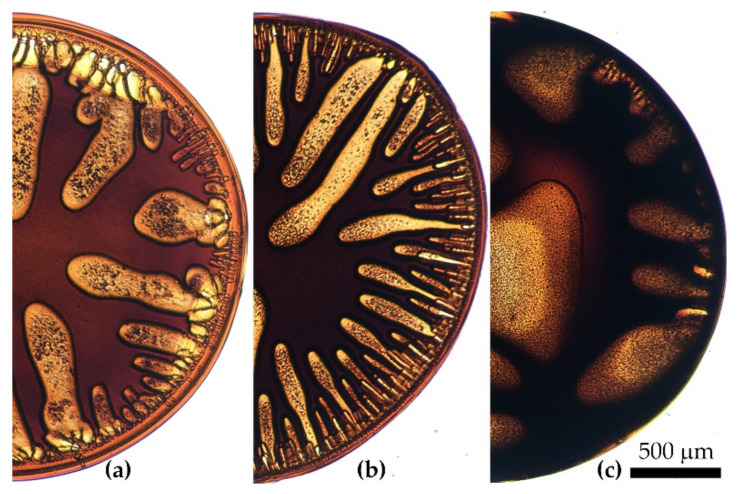
Coagulated drops of a PANI-O solution with alcohols using the example of P2 (**a**)—ethanol, (**b**)—propanol, (**c**)—heptanol.

**Figure 10 materials-15-00808-f010:**
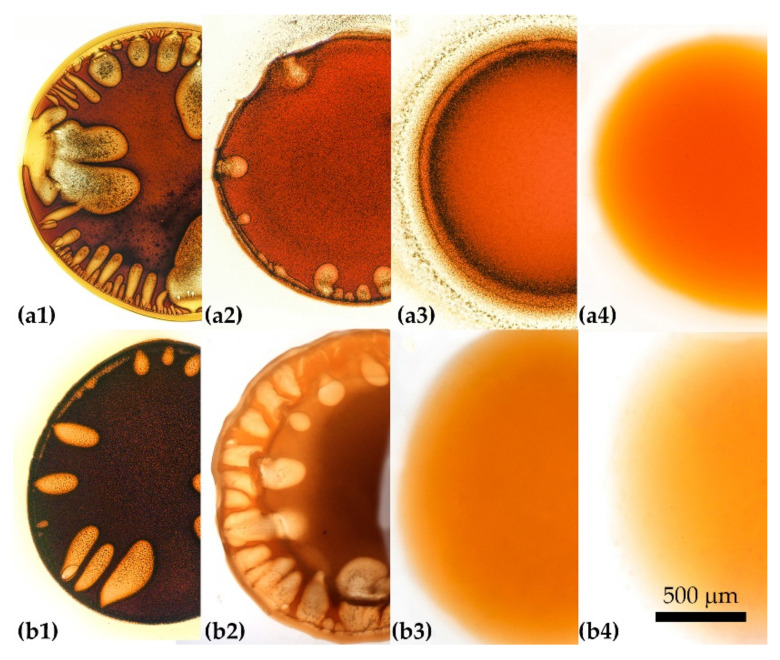
Coagulation of a PANI-O solution by two-component NMP-water (**a1**—**a4**) and NMP-ethanol (**b1**—**b4**) coagulants using the example of P2. Water content: (**a1**)—30, (**a2**)—20, (**a3**)—17, and (**a4**)—15%. Ethanol content: (**b1**)—50, (**b2**)—40, (**b3**)—30, and (**b4**)—20%.

**Figure 11 materials-15-00808-f011:**
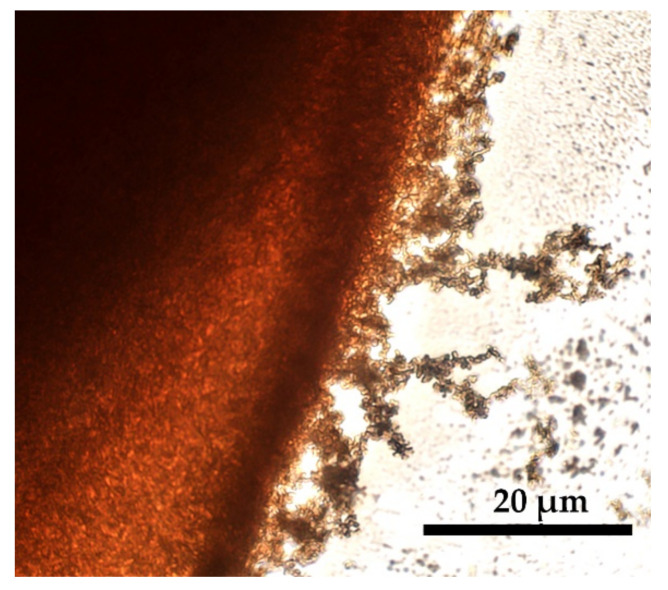
The image of PANI-O solution drop in contact with a soft coagulant (NMP 83% NMP + 17% H_2_O) causing selective disintegration and further partial dissolution.

**Figure 12 materials-15-00808-f012:**
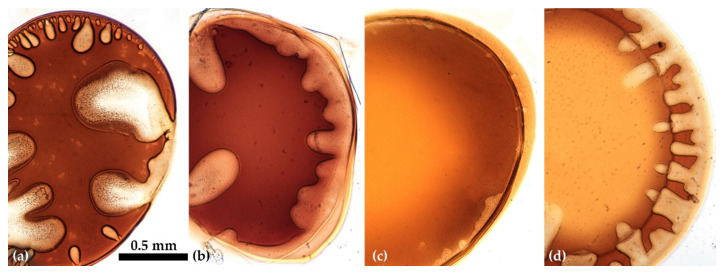
Images of the coagulated drops of a PANI-O solution with three-component compositions using the example of P4 consisting of ethanol/water/NMP of the following ratios: (**a**)—13/22/65, (**b**)—12/18/70, (**c**)—20/10/70, (**d**)—30/10/60.

**Figure 13 materials-15-00808-f013:**
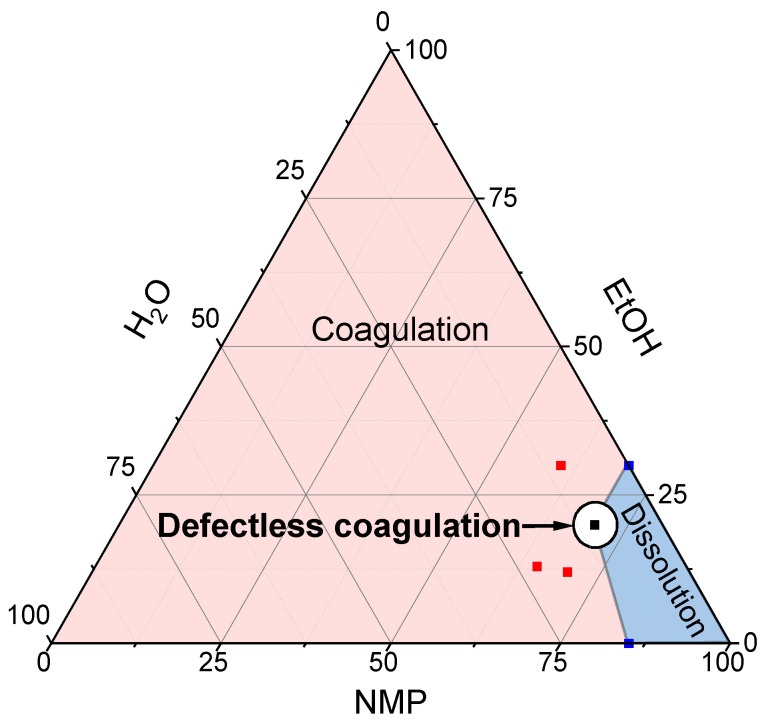
The diagram of coagulation-dissolution regions for the three-component coagulant. Red points indicate experimentally tested compositions.

**Figure 14 materials-15-00808-f014:**
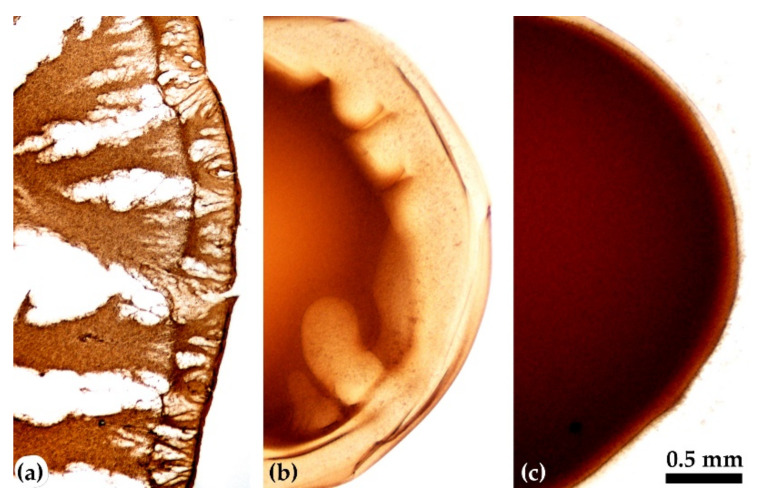
Coagulated drops of the sample P5 with concentrations: (**a**)—6, (**b**)—9, and (**c**)—12%. The coagulant is a water/NMP mixture (20/80).

**Figure 15 materials-15-00808-f015:**
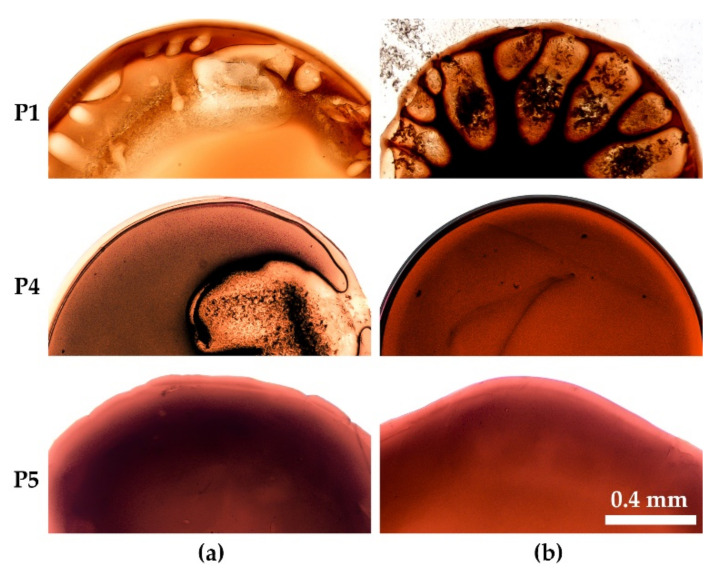
Coagulated drops of solutions P1, P4, P5 containing polymers with an intrinsic viscosity of 0.6, 1.2, and 1.8 dL/g, respectively (from up to down), by the following coagulants: (**a**)—water/NMP 20/80, (**b**)—ethanol/water/NMP 20/10/70.

**Figure 16 materials-15-00808-f016:**
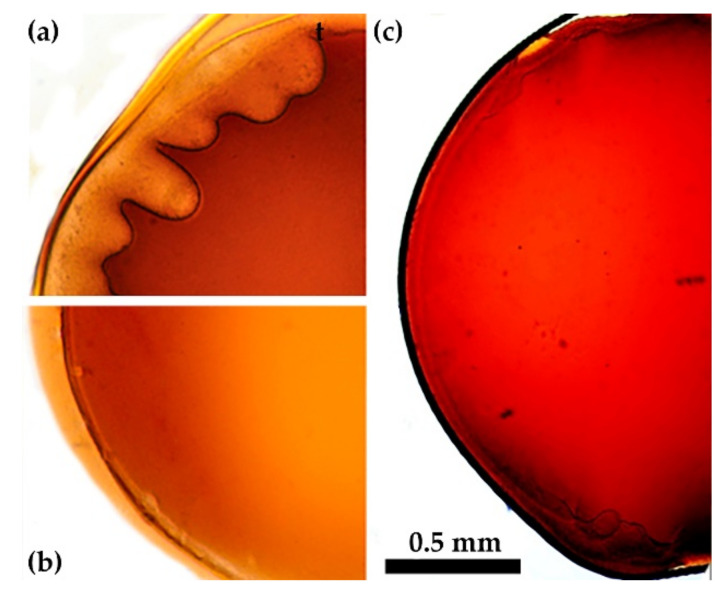
The coagulated drops of the spinning solutions P2 (**a**,**b**) and P5 (**c**) by three-component ethanol/water/NMP compositions in the ratio: 25/10/65 (**a**) 20/10/70 (**b**,**c**).

**Figure 17 materials-15-00808-f017:**
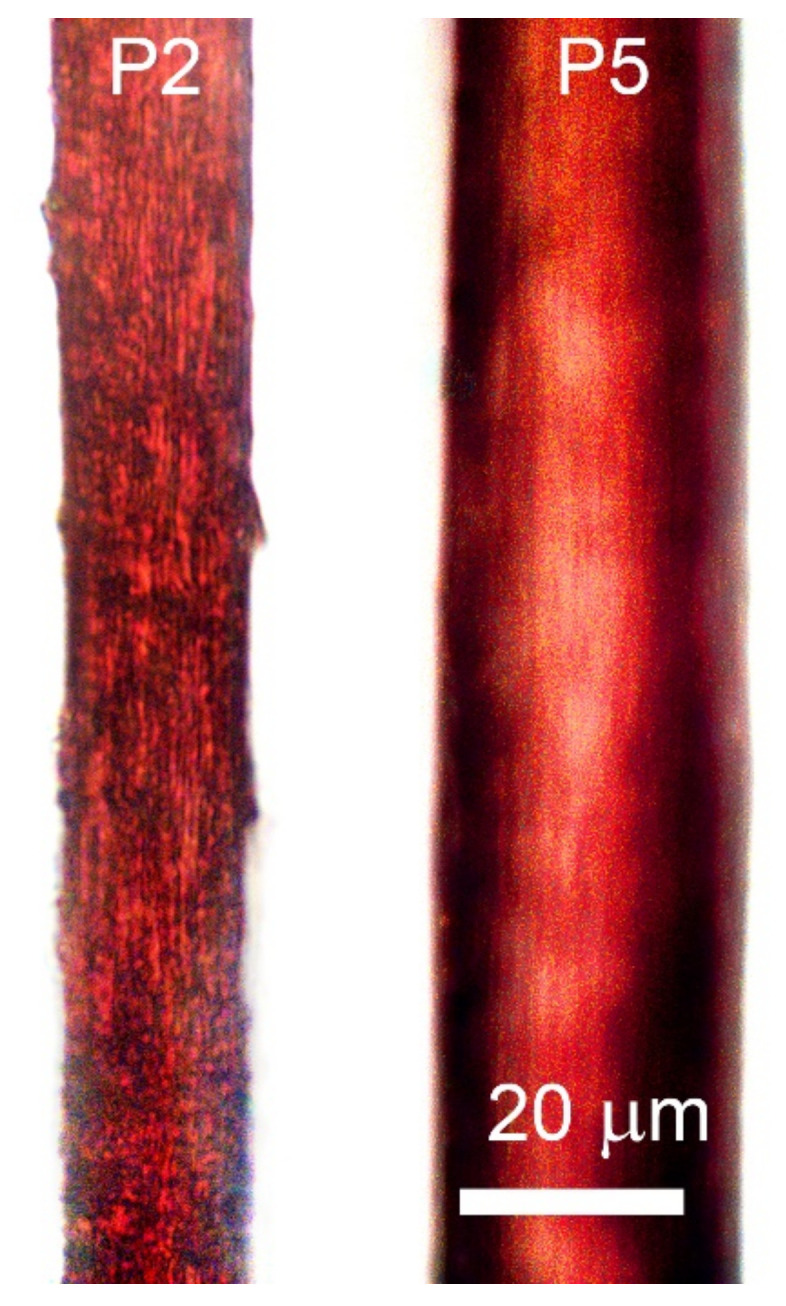
Images of the fibers obtained from PANI-O solutions P2 and P5, coagulated by three-component composition.

**Table 1 materials-15-00808-t001:** Characteristics of polymers and their solutions.

Sample	C^1^, % wt	[η], dL/g	Mv^2^, kg/mol
P1	10	0.6	16
P2	10	0.9	25
P3	15	0.9	25
P4	10	1.2	34
P5	10	1.8	54

^1^ Concentration of solutions obtained during the synthesis. ^2^ Calculated using the Mark–Kuhn–Houwink equation with the constants taken from [22].

**Table 2 materials-15-00808-t002:** Mechanical properties of spun fibers.

Sample	Strength, MPa	Elongation at Break, %	Modulus of elasticity, GPa	Diameter, µm	Total Draw Ratio
**P2**	144 ± 20	34 ± 5	1.9 ± 0.3	18 ± 2	1.2
**P5**	240 ± 31	50 ± 5	2.1 ± 0.2	27 ± 1	2.0

## Data Availability

All the data is available within the manuscript.

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
