# Peer review of "Some Specifics of Defect-Free Poly-(o-aminophenylene)naphthoylenimide Fibers Preparation by Wet Spinning"

_materials, 2022, doi:10.3390/ma15030808_

Round 1

Reviewer 1 Report

>Line 140, There should be „mm3” instead of „mm3”. Please check.

>Lines 232-243, Please explain why the P4-solution has been selected for this studies? Have the solutions of other polymers shown in Table 1 also been tested?

>Line 278, There is a double space.

>Line 309. Please explain why the P3-solution has been showed? What with the others polymers?

>Lines 367-372. Please explain why P4-solution has been tested?

>I don't understand why individual different polymers (once is P4 and once is P3) are shown in Sections 3.2.1; 3.2.1.1.; 3.2.1.2; 3.2.1.3. Perhaps it would be better to choose one polymer with the same parameters and show the influence of the parameters of the coagulant (s). Especially that in later subsections the influence of the concentration and molecular weight of the polymer is discussed.

>Line 450. Please remove “an”.

>Section 3 should be named “Results and Discussion”.

> Please justify, based on the research carried out, the selection of polymers P1 and P5 for research presented in section 4.

Author Response

Response to Reviewer 1 Comments

Point 1: >Line 140, There should be „mm3” instead of „mm3”. Please check.

Response: Thanks for noticing. That is corrected.

Point 2: >Lines 232-243, Please explain why the P4-solution has been selected for this studies? Have the solutions of other polymers shown in Table 1 also been tested?

Response: All samples were tested. More than a hundred experiments on the coagulation of solutions of different concentrations and molecular weight of the polymer by coagulants of various compositions have been performed. The majority of them were not included in the final version of the manuscript. Solution P4 was chosen as one of the characteristic systems in which both coagulation mechanisms are visible, namely, the growth of vacuoles with fast convection transfer of the coagulant and the diffusion mechanism. (Example is on the Figure SF1 in the Supplementary)

Point 3: >Line 278, There is a double space.

Response: Thanks for noticing. The extra space is deleted.

Point 4: >Line 309. Please explain why the P3-solution has been showed? What with the others polymers?

Response: Yes, of course, the other polymers have been investigated as well. In all cases, pure water leads to stiff coagulation with the defects, regardless of the viscoelastic properties / molecular weight of the polymer/concentration of the polymer solution. We have added a few examples in the Supplementary (Fig SF2-SF4).

Point 5: >Lines 367-372. Please explain why P4-solution has been tested?

Response: We have tested all the solutions. The sample P4 possesses the minimum sufficient parameters (molecular weight and solution viscoelastic properties) to implement defect-free coagulation. This has been shown in the schematic diagram of Fig. 13 and described in the lines 409-417

Point 6: >I don't understand why individual different polymers (once is P4 and once is P3) are shown in Sections 3.2.1; 3.2.1.1.; 3.2.1.2; 3.2.1.3. Perhaps it would be better to choose one polymer with the same parameters and show the influence of the parameters of the coagulant (s). Especially that in later subsections the influence of the concentration and molecular weight of the polymer is discussed.

Response: Thanks for the important question.

Indeed, choosing the one polymer with the same parameters looks properly.

We have conducted similar experiments with all samples. The final version of the Manuscript includes the most typical situations or the highest quality images in the case when there was no qualitative difference between the chosen solutions. For example, in Fig. 8, we could include any polymer instead of P3, but that sample is looked clearest and most suitable for illustrative aims. In Fig. 9, if we used the P1 solution, then there would be more defects in all cases, If, on the contrary, we used P5, then there would be fewer defects. Etc.

To avoid misunderstanding in the text of the article, the names of the figures have been changed for cases where the choice of a sample does not qualitatively affect the result obtained.

Point 7: >Line 450. Please remove “an”.

Response: Thanks for noticing. That is fixed.

Point 8: >Section 3 should be named “Results and Discussion”.

Response: Thanks for noticing. That is fixed.

Point 9: > Please justify, based on the research carried out, the selection of polymers P1 and P5 for research presented in section 4.

Response: I guess you meant P2 and P5.  Two solutions with the same concentration and twice different molecular weight of the polymer were selected based on the performed model experiments. Also, the most suitable three-component coagulant was selected based on the above studies, which allows obtaining defect-free samples. The P1 solution, judging by the preliminary data, was completely unsuitable for obtaining fibers (this is seen in Fig. 15). Honestly, we wanted to add this sample, but we couldn't even make the defective fiber from it. The selected P2 solution, judging by the coagulation experiments obtained, has almost sufficient properties to fiber spinning. Fig. 16 shows that it is necessary to use an even softer coagulant, which partially dissolves the as-spun fiber. From that solution, it turned out to make defective fiber. Solution P5 is the most optimal from the point of view of rheology and the possibility of using a three-component coagulant. This has been shown in our spinning experiment. The morphology of the resulting fibers and their mechanical properties correlate well with the expected results obtained from model coagulation experiments.

Reviewer 2 Report

  • “Three-component mixture containing solvent (NMP)”. It is wrong. Please find the complete word for NMP and write it in the abstract.
  • The position of Figures 1 and 2 is not suitable at the beginning of the introduction section. The authors may shift these two figures to the last part of the introduction.
  • Before performing the figures, the authors must write some brief about the study and the history of this research study. The introduction needs to rewrite.
  • Line 37, please avoid citing many references all together same with [2-7]. It is better to be a maximum of 3 references and no more. Please edit the citation in this section and replace the old references with new references.
  • Section 2 please separate “Materials” and “Methods”. Avoid writing melting temperature or preparation method in the material section.
  • Figure 3: it is not necessary to show the whole microscope setup. I recommend focusing just on the drop image and comparing the figure with other research studies.
  • The authors can use the following reference in this study:

Sabbagh, F., Kiarostami, K., Khatir, N. M., Rezania, S., Muhamad, I. I., & Hosseini, F. (2021). Effect of zinc content on structural, functional, morphological, and thermal properties of kappa-carrageenan/NaCMC nanocomposites. Polymer Testing93, 106922.

  • All the results must be compared with other similar studies' results.
  • Just as a suggestion the authors can add the future prospect for this research.

Author Response

Response to Reviewer 2 Comments

Point 1: “Three-component mixture containing solvent (NMP)”. It is wrong. Please find the complete word for NMP and write it in the abstract.

Response: Thanks for noticing. That is corrected.

Point 2: The position of Figures 1 and 2 is not suitable at the beginning of the introduction section. The authors may shift these two figures to the last part of the introduction.

Response:  Thanks for the suggestion. There are very few publications on PANI-O and PNBI polymers. Therefore, their structural formulas may be unfamiliar, so it was decided to place them in the top of introduction part where information about precursor and cyclized polymers is presented. Further, the published data on the rheological properties and morphology of PANI-O solutions and their coagulation are described, as well as the existing problems occurring during the fibers spinning.

Point 3: Before performing the figures, the authors must write some brief about the study and the history of this research study. The introduction needs to rewrite.

Response: Thanks for the suggestion. The introduction is expanded (lines 29-33):

Fire- and heat-resistant semi-ladder polynaphthoylbenzimidazole (PNBI) have one the highest thermal, heat-, fire resistance properties and limiting oxygen index among all organic polymers [2,3,7]. Application of that polymer is limited by the solubility only in strong acids. In addition,  the low solution concentration for fiber spinning or film casting determines the high cost of the product and makes the process ecologically unfriendly [6].

Point 4: Line 37, please avoid citing many references all together same with [2-7]. It is better to be a maximum of 3 references and no more. Please edit the citation in this section and replace the old references with new references.

Response: Thanks for noticing. The references were changed.

Point 5: Section 2 please separate “Materials” and “Methods”. Avoid writing melting temperature or preparation method in the material section.

Response: Thanks for the suggestion. We used the MDPI Template when we were writing the manuscript. The template specifies the use of the combined “Materials and Methods“ section.

Point 6: Figure 3: it is not necessary to show the whole microscope setup. I recommend focusing just on the drop image and comparing the figure with other research studies.

Response: Thanks for the suggestion. We have been successfully using the developed method for a long time [15, 8, 9, 23]. But unfortunately, its full description was used only in conference presentations. We periodically receive questions about a detailed description of the installation, and therefore it was decided to describe the method in as much detail as possible, and indicate that experiments can be carried on an ordinary microscope.

Point 7: The authors can use the following reference in this study:

Sabbagh, F., Kiarostami, K., Khatir, N. M., Rezania, S., Muhamad, I. I., & Hosseini, F. (2021). Effect of zinc content on structural, functional, morphological, and thermal properties of kappa-carrageenan/NaCMC nanocomposites. Polymer Testing, 93, 106922.

Response: Thanks for the suggestion. Unfortunately, the proposed article presents data on the structure and morphology of oxides in carrageenan hydrogels, which differs significantly from the topic of this study where coagulation conditions were investigated.

Point 8: All the results must be compared with other similar studies' results.

Response: According to our analysis of the open literature, only our group deals with PANI-O fibers spinning from organic solvents. At the same time, there were no systematic data on the influence of key parameters, such as the coagulant composition, polymer molecular weight, and solution viscosity on the coagulation process for wet spinning process as whole. That is why, this publication can be useful for other processes of wet spinning from polymer solutions.

Point 9: Just as a suggestion the authors can add the future prospect for this research.

Response:  Thanks for the suggestion. It was written in the introduction part (lines 88-94), Now that was extended:

  • to obtain high-quality heat-, fire resistance PNBI fibers with high limiting oxygen index from PANI-O solutions in NMP
  • This study is of particular interest for the development of membrane gas separation technology based on PNBI [20].

The authors prepared the PANI-O fibers by wet spinning through controlling coagulant and viscosities. The work is interestingly presented. These comments should be addressed before publication.

Reviewer 3 Report

The authors prepared the PANI-O fibers by wet spinning through controlling coagulant and viscosities. The work is interestingly presented. These comments should be addressed before publication.

  1. Authors are suggested to supply optical/digital images of their final products prepared under various conditions.
  2. How these fibers formed from the wet spinning are different from the fibers obtained from electrospinning in terms of physical properties? need to be discussed with related electrospinning-based works. doi.org/10.1080/00914037.2020.1857381; https://www.sciencedirect.com/science/article/pii/S002197971630128X
  3. how authors controlled the diffusion activity of the coagulant which is being said affecting to make defect-free fibers.
  4. are the images in Figure 17 are wet fibers or dried? need to show both.

Author Response

Response to Reviewer 3 Comments

Point 1: “Authors are suggested to supply optical/digital images of their final products prepared under various conditions.

Response: The paper presents the data of modeling the coagulation process for the selection of conditions for the traditional method of obtaining fibers from solutions by the wet method.

To compare the model experiments with the real process, two compositions were selected from which both defective (as predicted by the model experiment) and defect-free fibers were obtained. In this case, the strength of the defect-free fibers, as expected, turned out to be twice as high.

Point 2: How these fibers formed from the wet spinning are different from the fibers obtained from electrospinning in terms of physical properties? need to be discussed with related electrospinning-based works. doi.org/10.1080/00914037.2020.1857381; https://www.sciencedirect.com/science/article/pii/S002197971630128X

Response: It has not been possible to obtain fibers from PANI-O solutions using the needle free, Nanospider™ technology. We are also currently trying to select systems for dry or dry-wet spinning, but these experiments have not yet made it possible to obtain fibers.

Point 3: how authors controlled the diffusion activity of the coagulant which is being said affecting to make defect-free fibers.

Response: We control the diffusion activity by studying the kinetics of the coagulation process in a model experiments described in detail on the lines 130-165

Point 4: are the images in Figure 17 are wet fibers or dried? need to show both.

Response: These are images of dried fibers. As practice has shown, surface defects are often not visible on wet fibers due to the close refractive indices of the polymer and the solvent. Therefore, it is the dried fibers that we are examining. In addition, the study of mechanical properties requires the measurement of the diameter, which we do by measuring the samples in an optical microscope with maximum magnification for dry objectives, since diameter measurements are carried out precisely for analysis of extension tests.

Round 2

Reviewer 1 Report

Many thanks to the authors for their comprehensive responses to my comments. The paper is ready to be published in the journal.